# Intense PSMA Uptake in a Vertebral Hemangioma Mimicking a Solitary Bone Metastasis in the Primary Staging of Prostate Cancer via ^68^Ga-PSMA PET/CT

**DOI:** 10.3390/diagnostics13101730

**Published:** 2023-05-13

**Authors:** Farid Gossili, Clarissa G. Lyngby, Vibeke Løgager, Helle D. Zacho

**Affiliations:** 1Department of Nuclear Medicine and Clinical Cancer Research Center, Aalborg University Hospital, 9000 Aalborg, Denmark; h.zacho@rn.dk; 2Department of Clinical Medicine, Aalborg University, 9000 Aalborg, Denmark; 3Department of Radiology, Copenhagen University Hospital Herlev and Gentofte, 2730 Herlev, Denmark; c.gevargez@icloud.com (C.G.L.); vibeke.loegager@regionh.dk (V.L.)

**Keywords:** ^68^Ga-PSMA PET/CT, intense PSMA uptake, oligometastatic prostate cancer, atypical hemangioma, MRI, bone algorithm CT

## Abstract

A 78-year-old man with newly diagnosed high-risk prostate cancer underwent ^68^Ga-PSMA PET/CT for primary staging. This showed a single, very intense PSMA uptake in the vertebral body of Th2, without discrete morphological changes on low-dose CT. Thus, the patient was considered oligometastatic and underwent MRI of the spine for stereotactic radiotherapy planning. MRI demonstrated an atypical hemangioma in Th2. A bone algorithm CT scan confirmed the MRI findings. The treatment was changed, and the patient underwent a prostatectomy with no concomitant therapy. At three and six months after the prostatectomy, the patient had an unmeasurable PSA level, confirming the benign etiology of the lesion.

**Figure 1 diagnostics-13-01730-f001:**
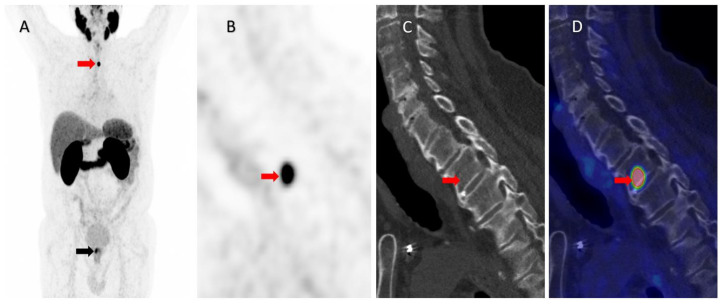
A 78-year-old patient with newly diagnosed high-risk prostate cancer (PCa), Gleason score 9 (4 + 5), PSA 14 ng/mL, and cT1c, underwent ^68^Ga-PSMA (Prostate-Specific Membrane Antigen) PET/CT for primary staging. ^68^Ga-PSMA PET showed, in addition to a high focal uptake in the prostate (SUVmax 11.9) (black arrow), an intense uptake in the second thoracic vertebra (Th2), as illustrated by the red arrow in the maximum intensity projection (**A**). A highly intense PSMA uptake (SUVmax 32.6) was observed in Th2 in the sagittal view of the PET image (**B**) with discrete but non-characteristic morphologic changes on the corresponding low-dose, nonenhanced CT (**C**) and fused PET/CT images (**D**). The lesion was highly suspicious of a solitary bone metastasis, and the patient was considered oligometastatic owing to the known high diagnostic accuracy for the detection of bone metastases with very few false-positive findings [1,2,3].

**Figure 2 diagnostics-13-01730-f002:**
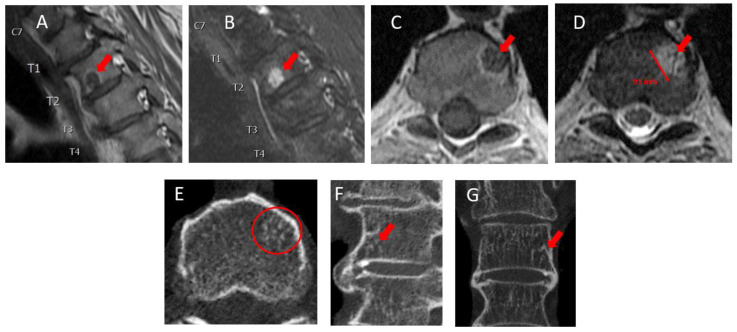
The patient underwent MRI of the thoracic spine for stereotactic radiotherapy planning. Sagittal T1-weighted (**A**), sagittal T2-weighted Dixon water (**B**), axial T1-weighted (**C**), and axial T2-weighted (**D**) non-contrast MRI images showed a well-defined, rounded lesion on the left side of the lower endplate of vertebra Th2, measuring 9 mm in diameter. The lesion was hyperintense on T2-weighted imaging and hypointense on T1-weighted imaging with a thin wall of high signal intensity around the lesion (arrows), which could be compatible with an atypical vertebral hemangioma (VH). While typical VHs present with high signal intensity in both T1- and T2-weighted sequences, atypical VHs are lipid-poor and have a greater vascular content. Therefore, they present with low signal intensity on T1-weighted images. Metastatic vertebral lesions may mimic the MR radiographic features of atypical VHs and are therefore considered as a main differential diagnosis [4,5]. To confirm the diagnosis of a VH, the patient underwent non-contrast CT using a bone algorithm with 6 mm reconstruction in a plane perpendicular to the Th2 vertebral body. This showed numerous small, calcified dots in the axial (**E**) plane (circle), known as the “polka-dot-sign”, which is a classic characteristic of VHs [5,6]. These changes often present with a vertically striated appearance in sagittal and coronal images, known as the “jail-bar” or “corduroy cloth” sign [6], but this feature was not clearly exhibited in this case. However, the sclerotic margin of the lesion in the sagittal (**F**) and coronal (**G**) images (arrows) is unlikely to be seen in a vertebral metastatic lesion. Consequently, the patient was deemed nonmetastatic and underwent robot-assisted radical prostatectomy with extended lymph node dissection, with no concomitant treatment. Histopathology demonstrated pT3apN0, Gleason score 9 (4 + 5), and ISUP 5 [7] PCa. At the 3-month and the 6-month follow-up, PSA was <0.1 ng/dl, confirming no distant metastases. PSMA uptake in a VH has previously been demonstrated [8,9,10]; however, in the published cases, either the PSMA uptake was lower (SUV 7.0) [9] or still low [10] or the MRI had the distinct radiographic features of a typical VH [8]. It is noteworthy that PSMA-avid osseous hemangiomas are uncommonly reported outside the vertebrae, with instances of biopsy-confirmed, intense uptake in iliac hemangioma [10] and mild uptake in calvarial hemangioma [11]. With the increasing use and guideline recommendation of PSMA PET/CT for the primary staging of PCa [12], the present case emphasizes the need for the careful evaluation and use of confirmatory imaging prior to the selection of treatment, including targeted treatment, for oligometastatic disease.

## Data Availability

This article comprehensively presents all relevant data pertaining to the subject under investigation. For any further inquiries or clarifications, the corresponding author can be contacted.

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
