# Peer review of "Intense PSMA Uptake in a Vertebral Hemangioma Mimicking a Solitary Bone Metastasis in the Primary Staging of Prostate Cancer via 68Ga-PSMA PET/CT"

_diagnostics, 2023, doi:10.3390/diagnostics13101730_

Round 1

Reviewer 1 Report

citation of this article as they talk about similar scenario in Figure 9 - Nonprostatic diseases on PSMA PET imaging: a spectrum of benign and malignant findings | Cancer Imaging | Full Text (biomedcentral.com)

There is no biopsy and follow-up to show stability; any of which would be useful to say that this is benign. But I think PSA of <0.1ng/mL is likely pointing to this. Maybe they can include 3-6months PSA trend if available. 

In the first line itself, they use short forms without explaining what is Th1 and PSMA

Author Response

Response to the editor and reviewers

Reviewer 1:

  1. citation of this article as they talk about similar scenario in Figure 9 - Nonprostatic diseases on PSMA PET imaging: a spectrum of benign and malignant findings | Cancer Imaging | Full Text (biomedcentral.com)

Answer: The mentioned study has been incorporated into the references section of the manuscript.

Changes in the manuscript: In order to reflect this change, the legend of figure 2 (located on page 2, lines 21-22) now includes the reference as number 10.

  1. There is no biopsy and follow-up to show stability; any of which would be useful to say that this is benign. But I think PSA of <0.1ng/mL is likely pointing to this. Maybe they can include 3-6months PSA trend if available.

Answer: Thank you for your comment. In cases where morphological findings are sufficient to establish a diagnosis, a biopsy is typically unnecessary. However, given that we currently have a PSA reading from the 6-month follow-up, we have incorporated the suggested change made by the reviewers.

Changes in the manuscript: As a result, modifications have been made to the legend of figure 2 (located on page 2, line 20) as well as the abstract (located on page 1, line 7) of the manuscript.

  1. In the first line itself, they use short forms without explaining what is Th1 and PSMA

Answer: The change in the manuscript has been implemented in accordance with the reviewer's recommendation.

Changes in the manuscript: To reflect this modification, specific details have been included in the legend of figure 1, as indicated on page 1, lines 2-4.

Reviewer 2 Report

Thank you for the opportunity to review this case report. There are at least 4 other similar case reports published previously.

The main comment is if there was any biopsy confirmation of this lesion. 

Other comments

-what was SUVmax of the prostate?

-did this patient commence ADT?

-what was the original proposed management to the primary? 

-once the authors were convinced that this was not a metastasis, was radiation to the prostate considered instead of radical prostatectomy?

-there are other references to osseous hemangiomas (PMID 33088371 & 

32169115) and it would also be worthwhile referencing PMID 31937920

-What was the Gleason score the radical prostatectomy?

-Do the authors have PSAs at longer follow up?

Author Response

Response to the editor and reviewers

Reviewer 2:

Thank you for the opportunity to review this case report. There are at least 4 other similar case reports published previously.

  1. The main comment is if there was any biopsy confirmation of this lesion.

Answer: We would like to express our gratitude for your feedback. In instances where there are adequate morphological observations to make a diagnosis, a biopsy is commonly deemed superfluous. Moreover, the spontaneous PSA drop to < 0.1ng/mL after prostatectomy with no concomitant systemic treatment implies that no metastases were present. We have added the information about unmeasurable PSA value at 6 months after prostatectomy (please see the answer to reviewer 1 above)

  1. What was SUVmax of the prostate?

Answer: Thanks for the comment. In accordance with the reviewer's suggestion, we have included the SUVmax of the prostate in our analysis.

Changes in the manuscript: As a direct result of this change, modifications have been made to the legend of figure 1, which is located on line 4 of page 1 within the manuscript.

  1. Did this patient commence ADT?

Answer: The patient received no systemic treatment (and thus no ADT) prior to, during or after prostatectomy.

  1. what was the original proposed management to the primary?

Answer: Upon initial diagnosis, the patient's condition was identified as T1cN0M1b disease, and the initial treatment plan included enrollment in the Oligometastatic protocol. This intervention protocol typically involves the performance of prostatectomy in tandem with stereotactic radiotherapy targeted towards a solitary metastatic site, followed by a six-month course of castration therapy. However, subsequent to the formulation of the initial treatment plan, a benign diagnosis of a single osseous lesion resulted in a change of approach. As a result, prostatectomy was performed as a standalone intervention, without any concomitant therapeutic interventions.

  1. once the authors were convinced that this was not a metastasis, was radiation to the prostate considered instead of radical prostatectomy?

Answer: Following the patient's assessment, curatively intended treatment was recommended, with options including radiation therapy or surgery. The patient was fully informed of the potential risks associated with both treatment approaches, and expressed a preference for prostatectomy as the preferred treatment modality.

  1. there are other references to osseous hemangiomas (PMID 33088371 & 32169115) and it would also be worthwhile referencing PMID 31937920

Answer: In response to the query, two of the studies (PMID 32169115 and 31937920) have been incorporated into the manuscript's references section. However, the study cited with PMID 33088371 pertains to calvaria hemangiomas, which is distinct from vertebral hemangioma, the focus of our discussion. Thus, this particular study has not been included in the document.

Changes in the manuscript: As for the changes made in the manuscript, it is worth noting that in order to reflect the mentioned changes, the figure legends have been updated. Specifically, the legend for Figure 1 (which is located on page 2, line 10) now includes the reference as number 3 (PMID 31937920). Similarly, the legend for Figure 2 (located on page 2, lines 21-22) now includes the reference as number 10 (PMID 32169115).

  1. What was the Gleason score the radical prostatectomy?

Answer: In response to the inquiry, it is important to note that the prostatectomy as previously stated in the manuscript, was classified as ISUP 5 PCa, with a specific Gleason score of 9 (4+5).

Changes in the manuscript:  Regarding the modifications made in the manuscript, it should be highlighted that the legend of figure 2, which can be found on page 2 at line 20, has been updated to reflect this information.

  1. Do the authors have PSAs at longer follow up?

Answer: Due to the current availability of PSA data obtained from a six-month follow-up, we have taken into consideration the recommendations provided by the reviewer and incorporated these observations in our analysis.

Changes in the manuscript: As a direct consequence of the recommended revision, alterations have been applied to the legend of figure 2, which is situated on line 20 of page 2, and the abstract, which is located on line 7 of page 1, within the manuscript.

Round 2

Reviewer 2 Report

Thank you for second opportunity to review and the improvements made in the manuscript.

1. It would be worthwhile stating if the other papers on hemangiomas had confirmed biopsies.

2. Due to the rarity of osseous hemangiomas, it think it would be important to reference PMID 33088371 and briefly mention the SUVmax etc.

3. The authors mention their "Oligometastatic protocol" in the reply which consists of radical prostatectomy, SABR to oligometastatic site and 6 months ADT. Can the authors please refer to the clinical trial that they were originally accruing patients to for this cohort? As the authors are aware, none of the guidelines (AUA, NCCN, EAU) recommend prostatectomy in this cohort whilst radiotherapy is a potential option in low-volume disease.

Author Response

Response to the reviewer – round 2

Reviewer 2:

  1. It would be worthwhile stating if the other papers on hemangiomas had confirmed biopsies.

Answer: We appreciate your valuable comment, which has been duly incorporated into the manuscript.

Changes in the manuscript: As a result, the legend for Figure 2 (positioned on page 2, lines 59) has been revised to reflect this modification.

  1. Due to the rarity of osseous hemangiomas, it think it would be important to reference PMID 33088371 and briefly mention the SUVmax etc.

Answer: We appreciate your insightful comment, which has been duly addressed by incorporating the pertinent reference into the manuscript alongside other osseous hemangiomas.

Changes in the manuscript: To reflect this modification, the legend for Figure 2 (situated on page 2, lines 57-59) has been updated to include the reference in question, along with a brief explanation.

  1. The authors mention their "Oligometastatic protocol" in the reply which consists of radical prostatectomy, SABR to oligometastatic site and 6 months ADT. Can the authors please refer to the clinical trial that they were originally accruing patients to for this cohort? As the authors are aware, none of the guidelines (AUA, NCCN, EAU) recommend prostatectomy in this cohort whilst radiotherapy is a potential option in low-volume disease.

Answer: We extend our gratitude for your meticulous review of our manuscript. The patient was referred for inclusion in the OLIGOMET-DK trial, which is registered at clinicaltrials.gov (ID NCT04086290). This trial includes prostatectomy, SABR to oligometastatic sites, and 6 months of ADT.

Round 3

Reviewer 2 Report

Thank you again for the opportunity to review. Good luck for your team in the future!